# A Radiographic Analysis of Proximal Humeral Anatomy in Patients with Primary Glenohumeral Arthritis and Implications for Press-Fit Stem Length

**DOI:** 10.3390/jcm11102867

**Published:** 2022-05-19

**Authors:** Easton J. Bents, Brian C. Werner, Justin W. Griffin, Patric Raiss, Patrick J. Denard

**Affiliations:** 1Oregon Shoulder Institute, 2780 E. Barnett Road, Suite 200, Medford, OR 97504, USA; eastongp@gmail.com; 2University of Virginia, 45 Ray C Hunt Dr Suite 1100, Charlottesville, VA 22903, USA; bcw4x@hscmail.mcc.virginia.ed u; 3Jordan-Young Institute, 5716 Cleveland Street, Suite 200, Virginia Beach, VA 23462, USA; justinwilliamgriffin@gmail.com; 4Orthopädische Klinik München, Steinerstrasse 6, 81369 Munich, Germany; patric.raiss@gmail.com

**Keywords:** total shoulder arthroplasty, glenohumeral arthritis, humeral stem, arthroplasty, stem length, shoulder replacement

## Abstract

While short stems in total shoulder arthroplasty (TSA) preserve bone stock and facilitate revision surgery, they have been associated with higher rates of malalignment and loosening in some cases compared to standard length stems. The purpose of this study was to analyze the intramedullary canal in progressive increments distal to the greater tuberosity to provide anatomic information about the optimal length of press-fit short stems for alignment and stability in TSA. We hypothesized that the humeral canal diameter will remain variable for the first 50 to 75 mm distal to the greater tuberosity and will become consistent thereafter. A retrospective review of 99 consecutive patients undergoing TSA with CT scans was performed. Intramedullary anterior-posterior (AP) and medial-lateral (ML) width as well as diameter were analyzed on two-dimensional computed tomography following multiplanar reconstruction. Measurements were taken at consistent distances distal to the greater tuberosity (GT). The transition point was measured at the proximal level of the humerus where endosteal borders of the medial and lateral cortices became parallel. The mean transition point was 73 mm from the GT (range: 53 to 109 mm). ML and AP widths became consistent 80 mm distal to the GT. IM diameter became consistent after 90 mm distal to the GT and a stem length of 90 mm extended past the transition point in 91.9% of cases. In TSA, a humeral stem length of 90 mm is required to predictably reach points at which the humeral canal becomes cylindrical and consistent in diameter. This information may aid data-driven decisions on humeral stem length during press-fit fixation, assuring consistency of alignment and implant stability, while maintaining ease of revision associated with a short stem implant. Level of evidence: III

## 1. Introduction

Total shoulder arthroplasty (TSA) is an effective treatment to restore function and reduce pain with over 90% of patients having good to excellent outcomes in long-term follow-up studies with standard length humeral stems [1,2,3]. Anatomic TSA has increased in usage from 9.5 cases per 100,000 persons in 2012 to 12.5 cases per 100,000 in 2017, and reverse TSA has increased in usage from 7.3 cases per 100,000 persons in 2012 to 19.3 cases per 100,000 in 2017 [4]. However, the postoperative complication rate for revision shoulder arthroplasty has been shown to be high at 22% [5].

The potential need for revision led to the development and popularity of press-fit components and eventually short and/or convertible humeral components. Whereas standard length stems are 100 to 150 mm in length, short stems are typically 50 to 100 mm in length [6]. The goal of the shorter length is to preserve proximal humeral bone stock and decrease stress shielding. Finite element analysis as well as clinical studies have supported the concept that shorter stems can reduce stress shielding [6,7,8,9,10,11]. Additionally, revision data supports the concept that the operative time for the removal of short stems is reduced compared to standard length stems [12]. However, potential trade-offs of short stems may include malalignment and stem loosening.

Varus or valgus malalignment of short stems has been reported to range from 10% to 47% [13,14,15,16,17,18,19,20,21,22,23,24]. The rate of stem loosening with short stems has also been variable, but overall has suggested a higher rate of loosening with short stems compared to standard length stems. For example, Zmistowski et al. [25] reported that 18% of short stems loosened at an average follow-up of 1.5 years. Similarly, Denard et al. [26] reported higher rates of loosening with a short stem compared to a standard length stem at a minimum 5-year follow-up. While these findings of malalignment and loosening may vary by design, it is also possible that there is an optimal length of press-fit short stems based on anatomical measurements that could be used to improve alignment and decrease stem migration. Most commercially available short stems are 60 to 70 mm in length, which has been chosen by convention rather than scientific evidence.

The purpose of this study was to characterize the anatomy of the humeral canal in patients with primary glenohumeral arthritis. We hypothesized that the humeral canal diameter would remain variable for the first 50 to 75 mm distal to the greater tuberosity and would become consistent thereafter at a position we define as the transition point.

## 2. Materials and Methods

A retrospective review was performed on a consecutive series of patients undergoing TSA at 2 institutions between June 2020 and June 2021. Institutional review board approval was obtained prior to commencing the study. Inclusion criteria were: (1) primary arthroplasty, (2) a diagnosis of primary glenohumeral arthritis, and (3) availability of a preoperative computed tomography (CT) scan with 1 mm or less slice thickness performed for the purpose of preoperative glenoid component planning with capture of a minimum of the proximal 150 mm of the humerus. Exclusion criteria were: diagnoses other than primary arthritis (e.g., acute fracture, rotator cuff arthropathy, fracture sequalae), and revision surgery. Ninety-nine patients met the study criteria. The mean age was 67 years (age range, 38–88 years) and the cohort consisted of 58 males and 41 females.

### 2.1. Radiographic Evaluation

Radiographic measurements were performed by one author (EJB). The first 25 scans were reviewed together with measurements based on mutual agreement from a second author (PJD). Subsequent scans were then reviewed by one author (EJB). Radiographic measurements were analyzed using two-dimensional computed tomography with multiplanar reconstruction using Horos^®^, a free open-source medical image viewer (www.horosproject.org, accessed on 12 April 2022). In the first stage of assessment, shoulder CT scans were reconstructed using the 3D MPR module. Images were generated in three planes of view: coronal, sagittal, and axial (Figure 1 and Figure 2). Each series was formatted to view standard anterior-posterior (AP) position in all patients. The coronal and sagittal view axes were formatted so that the proximal-distal line (PD) bisected the center of the intramedullary (IM) canal (Figure 1). In turn, the computer program automatically formatted the origin of the AP and medial-lateral (ML) lines to the absolute center of the IM canal in the axial view. We define the term “transition point” as the distance from the GT down to the proximal level of the humeral diaphysis where the endosteal borders of the lateral and medial cortices become parallel within the IM (Figure 1A). Using the coronal view, Horos^®^ was utilized to measure and record the transition point for each patient.

Next, the distance between the most superior aspect of the greater tuberosity and the humeral metaphysis was recorded. The level of humeral metaphysis was defined as the most inferior aspect of the humeral head excluding the humeral osteophyte. The axes on the coronal view were then moved to this point (Figure 2A). Using the axial view, the ML and AP distance of the IM canal was recorded (Figure 2B). From this same view, the area of a perfect circle that best fit the interior of the IM canal without overlapping any bone was recorded as the inner diameter. Measurements were then repeated at 25 mm distal to the metaphysis and 50 mm distal to the metaphysis, followed by 10 mm increments up to 120 mm (approximately 150 mm distal to the greater tuberosity) (Figure 2A).

### 2.2. Statistical Analysis

All data were recorded in Microsoft Excel (version 16.43, 20110804). Following raw data input, the IM canal diameter was calculated for their respective segment of the humerus using the following equation:$$\mathrm{diameter}=2\sqrt{\frac{\mathrm{area}}{\pi }}$$

In order to determine where the canal became a consistent diameter, *p*-values from *t*-tests were calculated using SPSS version 28 (IBM, Armonk, NY, USA) to compare each level of measurement to the level above for all patients as a group and then separately by sex. This was carried out for the mean ML, AP, and IM diameter measurements at each level. To determine when the canal became cylindrical (as opposed to consistent diameter), *p*-values from *t*-tests where the mean absolute value of the difference between ML and AP measurements at each level were compared to the level above, where a difference of zero would represent a perfectly cylindrical segment. These calculations were also subsequently performed for both males and females and separately by sex. Lastly, proposed stems of 50, 60, 70, 80, 90, and 100 mm in length distal from the metaphysis were used to calculate the percentage of patients whose proposed stem lengths extended past their transition point. This percentage was recorded for total patients, male patients, and female patients at each proposed stem length.

## 3. Results

For the overall cohort, mean distance from the GT to the metaphysis was 30 ± 4.2 mm and the mean transition point was 73 ± 11.1 mm. For males, the mean distance from the GT to the metaphysis was 32 ± 3.8 mm (*p* < 0.001 vs. females) and the mean transition point was 78 ± 10.6 mm (*p* < 0.001 vs. females). For females, the mean distance from the GT to the metaphysis was 28 ± 3.3 mm (*p* < 0.001 vs. males) and the mean transition point was 66 ± 6.9 mm (*p* < 0.001 vs. males).

Canal measurements at each level for overall patients are summarized in Table 1. In comparing measurements at each level to the level above, there was no difference for both ML and AP widths starting at level 3 (*p* > 0.05) (Table 1). Similarly, for overall patients, the IM canal became closest to a perfect circle at level 3 (*p* = 0.041) (Figure 3). In other words, the canal became statistically cylindrical at 80 mm distal to the GT. Comparing IM diameter, there was no significant difference between levels 4 and 3 (*p* = 0.631) (Figure 4). In other words, the canal became relatively consistent in diameter after 90 mm distal to the GT.

### 3.1. Male vs Female

For males, there was no difference for both ML and AP widths starting at level 3 or 92 mm distal to the GT (*p* > 0.05) (Table 2). Similarly, there was no difference in IM diameter after level 3, which corresponded to 92 mm distal to the GT (*p* > 0.05) (Table 2). For females, there was no difference in ML, AP, and IM diameter after level 3, which corresponded to 88 mm distal to the GT (*p* > 0.05) (Table 3). Significant differences between males and females were identified throughout the proximal humerus in comparing IM diameter (Table 4).

### 3.2. Stem Analysis

In stem length analysis, no stem under 60 mm would have reached the transition point of any patient. A stem length of 60 mm would have reached the transition point in 8.1% of overall patients, including 1.7% of males and 17.1% of females. A stem length of 70 mm would have reached the transition point in 53.5% of overall patients, including 24.1% of males and 78.0% of females. A stem length of 80 mm reached the transition point in 76.8% of overall patients, including 65.5% of males and 97.6% of females. A stem length of 90 mm would have reached the transition point in 91.9% of overall patients, including 87.9% of males and 100% of females. A stem length of 100 mm reached past the transition point in 94.9% of the overall patients and 94.8% of males. Five male patients had a transition point of greater than 100 mm.

## 4. Discussion

The primary findings of this study were that, based on the mean “transition point” and intramedullary measurements, humeral canal dimensions become relatively consistent at 73 to 90 mm distal to the GT. Furthermore, the transition to a cylindrical canal occurs at a shorter distance in females compared to males. These anatomic findings may have important implications for stem length in shoulder arthroplasty.

While an impact on functional outcome is not proven, one drawback of short stems is the potential for component malalignment [14,15,16,17,19,21,22,23,24,27]. Denard et al. [16] demonstrated that 14% of patients with short stem implants (ranging in length from 60 to 65 mm) were noted to be placed in greater than 5° of varus or valgus. Similarly, in a retrospective review of 49 short stems ranging in length from 54 to 66 mm, Jost et al. [18] reported a greater than 10% incidence of varus stem placement. Multiple studies also report that extreme varus malalignment may result in the loss of humeral offset and tuberosity overhang [13,20,27]. Conversely, varus or valgus malalignment is rarely reported with a standard length stem. Denard et al. [28] reported that only 2% of standard length stems (ranging in length from 115 to 151 mm) were placed in varus or valgus. In our analysis the IM canal tapered to a “transition point” at an average of 73 mm distal to the GT, and in examining potential stem lengths, a stem length of 90 mm was necessary to extend past the transition point of over 90% of patients. Statistically, there was no difference in ML/AP width and canal diameter after approximately 80 mm distal to the GT, meaning the canal became cylindrical at this point. Finally, there was no difference in canal diameters between 80 to 90 mm distal to the GT. Therefore, a stem length of at least 90 mm is likely to reach a point in the humeral canal where the bone interface becomes homogenous and varus/valgus malalignment of the humeral component may be avoided. More studies comparing stem lengths are required, however, as to determine such a length would improve the stability and positioning of short stems.

In a finite element analysis, Razfar et al. [29] demonstrated that a potential advantage of reducing stem length is that cortical stresses better mimic the intact humerus. In this study the short stem length modeled was 50 mm long. Clinically, Denard et al. [28] demonstrated lower rates of stress shielding with a short stem implant compared to standard length stems for TSA. However, clinical loosening still remains a concern. In a radiographic evaluation of 73 press-fit short stems (66–98 mm) Casagrande et al. [15] reported a humeral loosening rate of 11%. Another study on the same stem reported a 21% risk for loosening, but noted only a 3% risk with the addition of proximal coating [21]. Standard length stems, on the other hand, may have lower rates of loosening. In a systematic review of 20 articles with cohort sizes ranging from 20 to 131 shoulders (1309 TSAs), the rate of stem revision for humeral loosening for standard length stems was 0% to 6% [17]. Therefore, the ease of revision and bone preservation should be weighed against the risk of revision based on stem length. Interestingly, little published information to date has been obtained to determine the ideal length of a short stem implant. Rather, lengths of short stems were selected based on convention.

To the best of our knowledge, there appears to be only one biomechanical study comparing implant length and stability in the shoulder where Diaz et al. [30] demonstrated that in reverse shoulder arthroplasty, shorter stem lengths are equally rotationally stable when compared to standard stem lengths but that standard stem lengths may allow for better canal alignment and improved stability due to the combination of diaphyseal engagement and metaphyseal fitting. Data from hip arthroplasty have demonstrated that motion at the bone–implant interface may prevent bone ingrowth resulting in the formation of fibrous tissue, which in turn causes loosening of the implant [31]. In a finite element analysis model of the hip, Reimeringer et al. [32] showed that micromotion increases with decreasing stem length. There is potential to extrapolate this concept to the humeral canal and then make recommendations for stem length based on anatomical measurements. We suggest that extending stem length to the point where anatomical bone dimension becomes consistent (approximately 90 mm) may provide greater resistance to migration. This is not to suggest that diaphyseal fill is the goal, but rather to suggest that having the tip of the stem pass the transition point is likely to provide greater resistance to bending moments. Complete diaphyseal fit should not be the goal in most cases because it has been shown that increased cortical contact with the stem increases radiographic changes in the bone [22]. Thus, it is important to simultaneously consider that extensive filling of the canal, and particularly diaphyseal fixation, can lead to proximal stress shielding.

Slight differences were observed between males and females in the current study. As expected, canal diameters were on average larger in males than females. Likewise, the GT to metaphyseal distance was 5 mm shorter and the transition point was 12 mm lower in females compared to males. However, the level at which canal dimensions became consistent was at level 3 in both sexes. Based on these data, it would appear that a consistent stem length with variable diameters would be most anatomic as opposed to having stem lengths increase with progressively larger diameters. However, further study would be required accounting for patient height before making definite conclusions.

There are several limitations to this study. The data in this study are limited to anatomic measurements and the implications about stem design are therefore not yet clinically supported. Biomechanical testing would be helpful in determining if stem length extending past the transition point decreases loosening. Furthermore, this study does not take into account the spatial position or curvature in the humeral canal. Future 3D modeling studies may be useful in the assessment of the proximal humerus in regard to optimal stem shape and length. Additionally, the implications are limited to press-fit fixation with a humeral stem and do not account for bone quality. Alternative approaches include cemented fixation or stemless designs. In the latter, varus/valgus alignment is determined by the humeral head cut itself and fixation is completely based on the proximal metaphyseal fixation. Lastly, *t*-test samples were samples of opportunity without any predicate analysis; thus, a power analysis was not relevant and therefore runs the risk that the study is underpowered.

## 5. Conclusions

In TSA, a humeral stem length of 90 mm is required to predictably reach points at which the humeral canal becomes cylindrical and consistent in diameter. This information may aid data-driven decisions on humeral stem length, assuring consistency of alignment and implant stability while maintaining the ease of revision associated with a short stem implant.

## Figures and Tables

**Figure 1 jcm-11-02867-f001:**
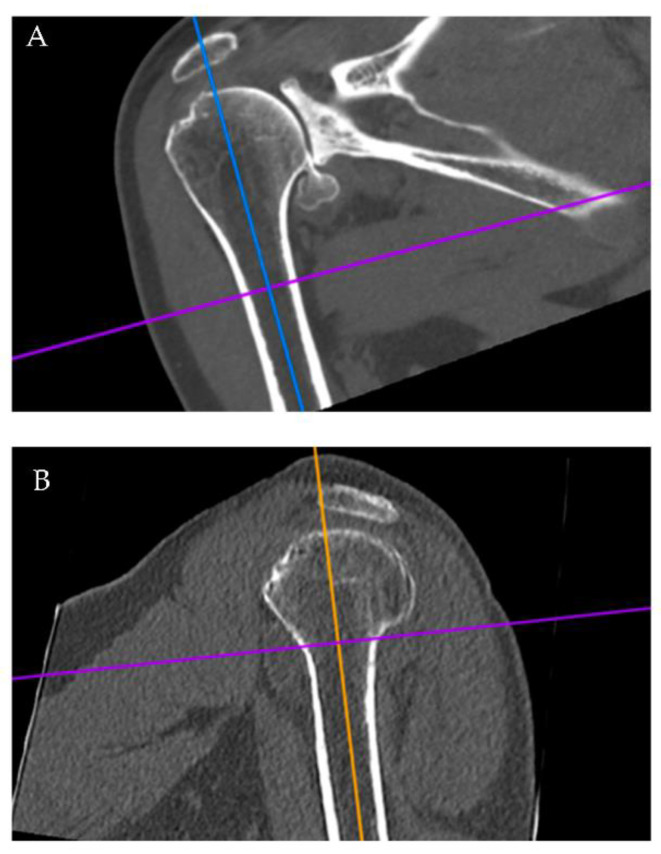
Initial two-dimensional multiplanar views for humeral measurements. The frontal view (**A**) depicts the PD line (solid blue) bisecting the IM canal and the transition point (solid purple line) where the diaphysis becomes parallel. The sagittal view (**B**) shows the PD line (orange line) bisecting the IM canal.

**Figure 2 jcm-11-02867-f002:**
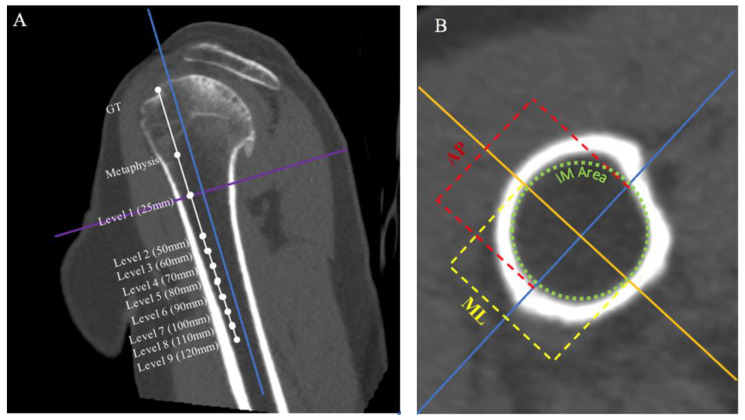
Two-dimensional-CT radiographic views depicting the process of measurement technique. Frontal view (**A**) of segmented humerus demonstrates measurements: greater tuberosity (GT) to metaphysis, Level 1—metaphysis, Level 2—25 mm distal to the metaphysis, Level 3—50 mm distal to the metaphysis, and then succeeding levels of 10 mm segments up to 120 mm distal to the metaphysis. The ML line (purple solid) depicts the cross section where measurements will be taken from the axial view and the PD line (blue solid) formatted to bisect the intramedullary (IM) canal. The axial view (**B**) depicts example measurements from the respective 25 mm segment from the axial view; ML axis (orange solid), AP axis (blue solid), ML (yellow dotted), AP (red dotted), and IM area (inside green dotted).

**Figure 3 jcm-11-02867-f003:**
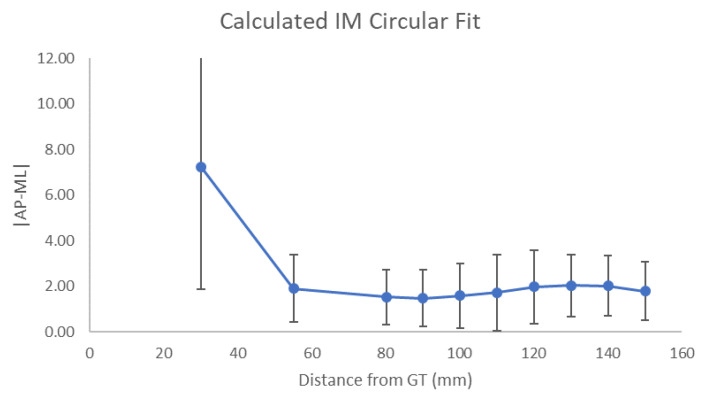
Calculated intramedullary (IM) canal closest fit to a perfect circle. Mean of absolute values (±standard deviation) of the difference between medial-lateral (ML) and anterior-posterior (AP) measurements taken at each level where the value of 0 represents a perfect circle. GT, greater tuberosity; mm, millimeters.

**Figure 4 jcm-11-02867-f004:**
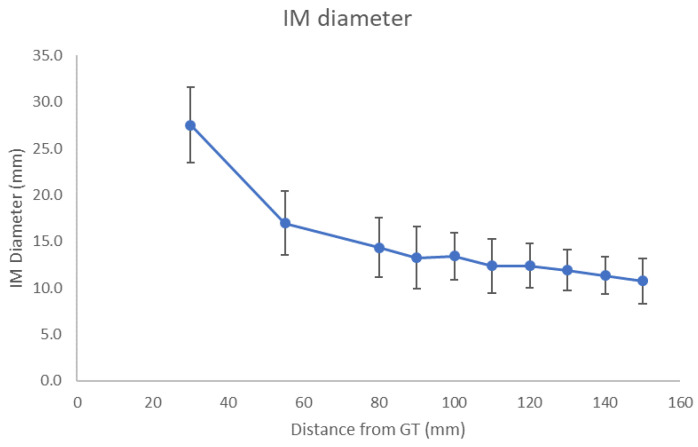
Intramedullary (IM) canal diameter. Mean values (±standard deviation) of diameter calculated from the area of best fit circles taken at each level. GT, greater tuberosity; mm, millimeters.

**Table 1 jcm-11-02867-t001:** Summarized data for overall patients.

Overall						
Total Distance From GT	ML (mm)	* p * -Value vs. Level above	AP (mm)	* p * -Value vs. Level above	IM Diameter (mm)	* p * -Value vs. Level above
**Metaphysis**	34.7		30.7		27.5	
**Level 1**	19.1	*<0.001*	18.4	*<0.001*	16.9	*<0.001*
**Level 2**	15.8	*<0.001*	15.5	*<0.001*	14.3	*<0.001*
**Level 3**	15.0	*0.058*	14.8	*0.075*	13.2	*0.018*
**Level 4**	14.4	*0.140*	14.3	*0.178*	13.4	*0.631*
**Level 5**	13.6	*>0.05*	13.7	*>0.05*	12.4	*>0.05*
**Level 6**	13.6	*>0.05*	13.5	*>0.05*	12.3	*>0.05*
**Level 7**	12.9	*>0.05*	13.1	*>0.05*	11.9	*>0.05*
**Level 8**	12.4	*>0.05*	12.5	*>0.05*	11.3	*>0.05*
**Level 9**	12.2	*>0.05*	12.5	*>0.05*	10.7	*>0.05*

GT, greater tuberosity; ML, medial-lateral; AP, anterior-posterior measurements; IM, intramedullary; mm, millimeters.

**Table 2 jcm-11-02867-t002:** Summarized data for males.

Males						
Total Distance From GT	ML (mm)	* p * -Value vs. Level above	AP (mm)	* p * -Value vs. Level above	IM Diameter (mm)	* p * -Value vs. Level above
**Metaphysis**	36.5		32.0		29.1	
**Level 1**	20.5	*<0.001*	19.7	*<0.001*	18.1	*<0.001*
**Level 2**	17.1	*<0.001*	16.6	*<0.001*	15.8	*<0.001*
**Level 3**	16.2	*0.065*	15.7	*0.070*	14.3	*0.005*
**Level 4**	15.5	*0.142*	15.1	*>0.05*	14.3	*>0.05*
**Level 5**	14.4	*>0.05*	14.4	*>0.05*	13.1	*>0.05*
**Level 6**	14.4	*>0.05*	13.8	*>0.05*	13.0	*>0.05*
**Level 7**	13.5	*>0.05*	13.5	*>0.05*	12.5	*>0.05*
**Level 8**	13.0	*>0.05*	13.0	*>0.05*	11.9	*>0.05*
**Level 9**	12.7	*>0.05*	12.9	*>0.05*	11.4	*>0.05*

GT, greater tuberosity; ML, medial-lateral; AP, anterior-posterior measurements; IM, intramedullary; mm, millimeters.

**Table 3 jcm-11-02867-t003:** Summarized for females.

Females						
Total Distance From GT	ML (mm)	* p * -Value vs. Level above	AP (mm)	* p * -Value vs. Level above	IM Diameter (mm)	* p * -Value vs. Level above
**Metaphysis**	32.1		29.0		25.3	
**Level 1**	17.0	*<0.001*	16.5	*<0.001*	15.3	*<0.001*
**Level 2**	13.9	*<0.001*	14.0	*<0.001*	12.3	*<0.001*
**Level 3**	13.3	*0.281*	13.6	*0.435*	11.7	*0.361*
**Level 4**	12.7	*>0.05*	13.0	*>0.05*	11.9	*>0.05*
**Level 5**	12.3	*>0.05*	12.6	*>0.05*	11.2	*>0.05*
**Level 6**	12.2	*>0.05*	12.8	*>0.05*	11.2	*>0.05*
**Level 7**	11.7	*>0.05*	12.1	*>0.05*	10.7	*>0.05*
**Level 8**	10.8	*>0.05*	11.1	*>0.05*	9.9	*>0.05*
**Level 9**	10.6	*>0.05*	11.4	*>0.05*	8.8	*>0.05*

GT, greater tuberosity; ML, medial-lateral; AP, anterior-posterior measurements; IM, intramedullary; mm, millimeters.

**Table 4 jcm-11-02867-t004:** Comparison of IM diameter between males and females.

	Diameter			
Total Distance from GT	Male (mm)	Female (mm)	Mean Diff (mm)	*p*
Metaphysis	29.1	25.3	3.8	<0.001
Level 1	18.1	15.3	2.9	<0.001
Level 2	15.8	12.3	3.6	<0.001
Level 3	14.3	11.7	2.6	<0.001
Level 4	14.3	11.9	2.4	<0.001
Level 5	13.1	11.2	1.8	0.004
Level 6	13.0	11.2	1.8	0.001
Level 7	12.5	10.7	1.8	0.002
Level 8	11.9	9.9	2.0	0.002
Level 9	11.4	8.8	2.6	0.004

GT, greater tuberosity; mm, millimeter.

## Data Availability

Details regarding where data supporting reported results can be requested at the following e-mail address: pjdenard@gmail.com.

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
