# Peer review of "A Radiographic Analysis of Proximal Humeral Anatomy in Patients with Primary Glenohumeral Arthritis and Implications for Press-Fit Stem Length"

_jcm, 2022, doi:10.3390/jcm11102867_

Round 1

Reviewer 1 Report

The present study depicts interesting radiographic measurements considering the Implantation of a TSA. There are some minor issues listed below:

Abstract: "easy of revision" -> ease of revision?

Text:

"Using the coronal view, we define the term "transition point" as the distance from the GT down to the proximal level of the humeral diaphysis where the endosteal borders of the lateral and medial cortices become parallel within the IM (Figure 1A)." -> At first I thought you directly measured the transition point by eyesight or likewise, but you should point out, that your whole study aims at the measurement of the transition point.

"Next, the distance between the most superior aspect of the greater tuberosity and the humeral metaphysis was recorded." How did you define metaphysis?

l 38: "Anatomic TSA" -> Please also provide data for reverse TSA .

l 58: "it also possible" -> it is also possible

l 59: "potential for stem migration." -> decrease stem migration

l 95f: "(A) depicts the PD line (solid purple) bisecting the IM canal and the transition point (solid blue line) where the diaphysis becomes parallel." -> purple and blue seem to be mixed up

l 252: "large" -> larger?

Reviewer 2 Report

In this anatomic study, a quantitative description of the proximal humeral canal is provided. The topic is extremely interesting, as it provides guidance for the design of prosthetic humeral stems.

The introduction is well written and focuses readers' attention towards the study topic.

Methods are clearly described. Nevertheless, some points might be improved: 

  • in the measurements, distaces are measured relative to the humeral metaphysis. However, no precise definition is provided of the definition of the humeral metaphysis point.
  • on lines 77-78, it is stated that measurement agreement was verified by a second author; no inter-rater agreement is described in the statistic section nor outcomes are provided in the results section. How has agreement been evaluated?
  • a t-test is used to check the cylidrical shape of the humeral diafisis. Was a power analysis performed to evaluate the adequate sample size for this statistical test?

Provided results are plainly exposed.

The discussion is informative and clear.
